# A Novel Protein–Protein Interaction between RSK3 and IκBα and a New Binding Inhibitor That Suppresses Breast Cancer Tumorigenesis

**DOI:** 10.3390/cancers13122973

**Published:** 2021-06-14

**Authors:** Hee-Sub Yoon, Sung Hoon Choi, Jung-Hyun Park, Jin-Young Min, Ju-Yong Hyon, Yeji Yang, Sejin Jung, Jae-Young Kim, Nam Doo Kim, Ji Hoon Lee, Eun Hee Han, Sung-Gil Chi, Young-Ho Chung

**Affiliations:** 1Research Center for Bioconvergence Analysis, Korea Basic Science Institute (KBSI), Cheongju 28119, Korea; youns1115@kbsi.re.kr (H.-S.Y.); burnes@nate.com (S.H.C.); jhpark7608@hanmail.net (J.-H.P.); mjymjy123@kbsi.re.kr (J.-Y.M.); hjy1234@kbsi.re.kr (J.-Y.H.); yangyj@kbsi.re.kr (Y.Y.); 2Department of Life Sciences, Korea University, Seoul 02841, Korea; 3Yonsei Liver Center, Yonsei University College of Medicine, Seoul 03722, Korea; 4New Drug Development Center, Daegu Gyeoungbuk Medical Innovation Foundation DGMIF, Daegu 41061, Korea; sjjung@dgmif.re.kr (S.J.); namdoo@voronoibio.com (N.D.K.); jhlee@dgmif.re.kr (J.H.L.); 5Graduate School of Analytical Science and Technology (GRAST), Chungnam National University, Daejeon 34134, Korea; jaeyoungkim@cnu.ac.kr

**Keywords:** protein-protein interaction (PPI), cell-based unidentified protein interaction discovery (CUPID), breast cancer, RSK3 (RPS6KA2), IκBα, binding inhibition

## Abstract

**Simple Summary:**

Breast cancer is the most common carcinoma and the leading cause of cancer-related death among women worldwide. Many kinases play important roles in the tumorigenesis of various cancers. IκBα phosphorylation is important for the regulation of NF-κB activity and is linked to the regulation of tumorigenesis. However, the kinase signaling network regulating IκBα phosphorylation in the context of cancer is not well understood. Herein, we report that RSK3 (RPS6KA2) is a novel binding partner of IκBα. RSK3 induces IκBα phosphorylation and NF-κB activation. Further, a chemical screening approach identified an inhibitor of RSK3/IκBα binding that impairs RSK3-mediated IκBα phosphorylation and decreases breast cancer cell survival, proliferation, and migration.

**Abstract:**

Multiple cancer-related biological processes are mediated by protein-protein interactions (PPIs). Through interactions with a variety of factors, members of the ribosomal S6 kinase (RSK) family play roles in cell cycle progression and cell proliferation. In particular, RSK3 contributes to cancer viability, but the underlying mechanisms remain unknown. We performed a kinase library screen to find IκBα PPI binding partners and identified RSK3 as a novel IκBα binding partner using a cell-based distribution assay. In addition, we discovered a new PPI inhibitor using mammalian two-hybrid (MTH) analysis. We assessed the antitumor effects of the new inhibitor using cell proliferation and colony formation assays and monitored the rate of cell death by FACS apoptosis assay. IκBα is phosphorylated by the active form of the RSK3 kinase. A small-molecule inhibitor that targets the RSK3/IκBα complex exhibited antitumor activity in breast cancer cells and increased their rate of apoptosis. RSK3 phosphorylation and RSK3/IκBα complex formation might be functionally important in breast tumorigenesis. The RSK3/IκBα-specific binding inhibitor identified in this study represents a lead compound for the development of new anticancer drugs.

## 1. Introduction

Breast cancer is the most common carcinoma and the leading cause of cancer-related death among women worldwide [1]. Because the molecular mechanisms underlying breast cancer have not been fully elucidated, identification of factors related to development of this cancer could provide insight into protein functions and potential therapeutic targets [2].

Protein–protein interactions (PPIs) are elementary biological processes; like differentiation, growth, and apoptosis, they must be mediated and controlled [3]. Multiple cancer-related processes are also mediated by PPIs. Therefore, PPIs are emerging as a promising new class of drug targets, and PPI inhibitors exert anticancer activity by blocking specific PPIs [3,4]. Although PPIs play essential roles in tumorigenesis, very few of these interactions have been identified as anticancer therapeutic targets.

The RSK (ribosomal S6 kinase) family is a group of highly related serine/threonine kinases that regulate diverse cellular processes, including cell survival, proliferation, growth, and motility [5]. The 90 kDa subfamily of RSK proteins has two discrete functional kinase domains: An N-terminal kinase domain (NTKD) and a C-terminal kinase domain (CTKD); the CTKD is essential for activation of the NTKD by the Ras-ERK1/2 pathways [5,6,7]. This family includes four isoforms: RSK1, RSK2, RSK3, and RSK4 [5,8], and these kinases have high sequence identity (75–80%) [5,9,10]. RSK2 is a tumorigenic factor [11,12], whereas RSK3 and RSK4 are tumor suppressors in ovarian cancer [13,14,15]. However, the mechanisms underlying the involvement of RSK3 and RSK4 in breast cancer are not clear [1,16,17,18]. Moreover, when the PI3K pathway inhibitors BEZ235 and BKM120 are administered to breast cancer cells, the levels of RSK3 and RSK4 increase. Moreover, it has been reported that PI3K pathway inhibitors (e.g., BEZ235 and BKM120) promote expression of RSK3 and RSK4, which is associated with drug resistance and tumorigenesis in breast cancer [16].

Recently, three different classes of RSK inhibitors were identified. BI-D1870 and SL0101 bind to the NTKD of RSK1, RSK2, and RSK4, and fluoromethylketone (FMK) binds to the CTKD of RSK1, RSK2, and RSK4; none of these inhibitors bind to RSK3 [5,7,19,20,21,22]. In particular, treatment of cells with SL0101 or RNAi against RSK1 or -2 inhibits the proliferation of human prostate and breast cancer cells, suggesting that these two isoforms positively regulate cancer cell proliferation [5,15,23]. Overexpression of RSK3 and RSK4 mediates resistance to the PI3K inhibitor BEZ235 in breast cancer cells [16]. Although RSK3 was initially thought to play positive roles in cell proliferation, its detailed functions and mechanisms are currently unknown [5,9,24].

NF-kappa B inhibitor alpha (IκBα) is a member of a family of cellular proteins that inhibit the transcription factor NF-κB [25], and IκBα phosphorylation is essential for regulation of NF-κB activity in the cell [4,26]. Serine 32 and 36 are permanent phosphorylation sites on IκBα that were identified based on activation of the IκB kinase complex [27]. Various factors affecting the expression of the IkB family have been identified in several cancers [26,28]. Interaction with the inhibitory IκB protein regulates NF-κB activity [29,30,31,32].

IκBα binding partners were identified by screening using the Cell-based Unidentified Protein Interaction Discovery (CUPID) system, which allows real-time monitoring of direct or indirect PPIs in living cells [3]. This system is based on the membrane trafficking translocation module of protein kinase C delta (PKCδ), which is translocated to the membrane from the cytosol after treatment with phorbol myristate acetate (PMA) [3].

In this study, we screened for novel PPI targets using a GFP (Green Fluorescence protein)-tagged kinase library to find IκBα binding partners and binding domains using CUPID analysis, with the intention of identifying PPI-specific inhibitors using mammalian two-hybrid (MTH) analysis. We hypothesize that these inhibitors could serve as lead compounds for anticancer drug development [33]. An IκBα binding partner was discovered that influences breast cancer cell survival and proliferation. This PPI-specific inhibitor we discovered decreased survival, proliferation, and increased apoptosis in breast cancer cells.

## 2. Materials and Methods

### 2.1. Materials

Phorbol Myristate Acetate (PMA) was obtained from MilliporeSigma (Merck, St. Louis, MO, USA). Fetal bovine serum (FBS) was produced by Welgene (Welgene Inc, Daegu, Republic of Korea), and penicillin–streptomycin solution and trypsin were obtained from HyClone (Hyclone Laboratories Inc, Logan, UT, USA). Recombinant protein [activated RSK3, IKK (IκB kinase) and inactive IκBα] was obtained from SignalChem (Signalchem Biotech Inc, Richmond, VA, Canada). IκBα antibody and phosphor-IκBα antibody were obtained from Cell Signaling Technology (Cell Signaling Technology, Beverly, MA, USA). Antibodies against RSK3, c-myc, GFP, and secondary antibodies were obtained from Santa Cruz Biotechnology (Santa Cruz Biotechnology, Dallas, TX, USA) and Turbofect and puromycin were obtained from Thermo Fisher Scientific (Thermo Fisher Scientific Inc, Waltham, MA, USA). CCK-8 kit is a product from Dojindo (Dojindo Laboratories, Kumamoto, Japan).

### 2.2. Cell Culture

Human breast cell lines (MDA-MB-231, MDA-MB-468, MCF7, and MCF10A) were taken from the American Type Culture Collection (ATCC, Manassas, VA, USA) and HEK293T cells were purchased from the Korean Cell Line Bank (KCLB, Seoul, Korea). All cells were maintained in DMEM (Dulbecco’s Modified Eagle Medium) (HyClone, Logan, UT, USA) supplemented with 10% FBS (Welgene Inc, Daegu, Republic of Korea), and 1% penicillin/streptomycin (Hyclone Laboratories Inc, Logan, UT, USA). All the cells used in this experiment were grown in a humidified chamber (37 °C, 5% CO_2_).

### 2.3. Cell Based Un-/Identified Protein Interaction Discovery (CUPID) Assay

HEK293T cells were grown on 96 well culture plate to 50−70% confluence. Transient co-transfection of the desired bait/prey plasmid pair (RSK3/IκBα) was performed using a Turbofect (Thermo Fisher Scientific Inc, Waltham, MA, USA). according to the manufacturer’s standard protocol. PMA was treated in the medium for 500 nM/well. Cell was fixed using 5% paraformaldehyde (Thermo Fisher Scientific Inc, Waltham, MA, USA) for 5 min and washed by PBS (HyClone, Logan, UT, USA) three times.

### 2.4. Confocal Laser Scanning Microscopy (CLSM)

For live cell imaging, transiently co-transfected cells on Poly-l-lysine-coated glass coverslips were mounted. Cells were put in the live cell chamber incubator (Chamlid HX, LCI, Seoul, Korea). This incubator was linked to the CO_2_ controllers and kept at 37 ℃ (Heating & Cooling system/CU-501 and Gas mixer/FC-5N, LCI, Seoul, Korea). The cells were washed with medium [DMEM without phenol red (HyClone, Logan, UT, USA) supplemented 10% FBS (Welgene Inc, Daegu, Republic of Korea), 37 °C] and then 1 mL of medium was added. Sequential images of the same cells were collected at 2 min intervals for 20 min using a laser scanning confocal microscope (LSM 710, Carl Zeiss, Germany) with a C-Apochromatic (LSM 710, Carl Zeiss, Germany) 40X/1.2 water immersion lens (488 nm argon laser/505–550 nm detection range for eGFP, 561 nm solid state laser/586–662 nm detection range for mRFP). During imaging, the total concentration of PMA in the medium was 500 nM and was added to the chamber.

### 2.5. ZOE Fluorescent Cell Imager

The ZOE Fluorescent Cell Imager (Bio-Rad, Catalog #1450031, Hercules, CA, USA) is an inverted imaging system with a brightfield and three fluorescent channels (emitting in blue, green, and red colors), suitable for routine cell culture and imaging applications. All channels are fully integrated and optimized for most commonly used fluorescent proteins and dyes.

### 2.6. Cell-Based ELISA of IκBα

IκBα activation was measured using a cell-based ELISA phospho-IκBα (S32/S36) assay kit (R&D Biosystems, 614 McKinley Pl NE, MN, USA) according to the manufacturer’s instructions. Briefly, HEK293T cells were seeded on the black 96-well plates (Thermo Fisher Scientific Inc, Waltham, MA, USA). RSK3 expression vector and shRNA (AGGTCCTGAAGCGTCAAGGCTATGATGCG) were transfected into HEK293T cells for 48 h. After that cells were fixed and incubated with primary and secondary antibody. After addition of the substrate, the fluorescence was measured with a microplate reader (SpectraMax M5, Molecular device, 3860 N First Street San Jose, CA 95134, USA).

### 2.7. Docking Model Prediction of RSK3/IκBα Interaction Inhibitor

To create a binding model of the RSK3/IκBα interaction inhibitor, the homology model structure of RSK3 was in silico docking with RSK3I. A homology model structure of RSK3 was built using the Prime module in Schrodinger package (Schrodinger, New York, NY, USA) and was further refined by using the OPLS (Optimized Potentials for Liquid Simulations) 2005 force field. For identifying potential RSK3I binding sites in RSK3, we used the Site Map tool in the Schrodinger Package (Schrodinger, NY, USA).

The RSK3I was built using Maestro build panel and minimized using the Macro model tool of Maestro in the Schrödinger package (Schrodinger, New York, NY, USA). The RSK3 homology model structure was minimized using the Protein Preparation Wizard by applying an OPLS2005 force field. Docking of an RSK3I molecule was performed using GLIDE (Schrodinger, New York, NY, USA). For the grid generation, the binding site was defined as the centroid of the Lys 534, Arg 531, Arg 585, Leu 623, and Pro 615 residues. The best-docked poses were selected as the lowest Glide score.

### 2.8. Mammalian Two Hybrid (MTH) Assay

The CheckMate MTH system (Promega, Madison, WI, USA) contains three expression vectors that termed pG5luc, pBIND, and pACT. The pBIND vector expresses a yeast GAL4 DNA-binding domain upstream of a multiple cloning region and a SV40-controlled renilla luciferase for transfection control. The pACT vector contains the *Herpes simplex* VP16 activation domain upstream of a multiple cloning region. The genetic information coding for the interactive proteins of interest (RSK3, IκBα) was subsequently cloned into the pBIND and pACT vectors to generate fusion proteins with the DNA-binding domain of GAL4 and the activation domain of VP16.

The GAL4 and VP16 fusion constructs (pBIND-IκBα, pACT-RSK3) were transfected in HEK293T cells. The MTH assay was performed as described by manufacturer protocol. The MTH assay was used to measure luciferase activity, which is an indicator of PPIs. The relative luciferase activity for pG5-luc was determined by normalizing firefly luciferase activity with *renilla* luciferase activity. Luciferase activity was measured using the Dual-Glo Luciferase Assay System kit (Promega, Madison, WI, USA) as specified by the manufacturer in an M4 molecular device spectrophotometer. Twenty-four hours after transfection, cells were subsequently washed once with phosphate-buffered saline (PBS). After addition of 200 μL of lysis buffer, cells were harvested and centrifuged (4 °C, 13,000 rpm, 5 min). Measurement was carried out in 1:1 dilutions of the cell extract with the Dual-Glo luciferase reagent (Promega, Madison, WI 53711 USA) followed by an incubation of 10 min within 2 h. All assays were performed in triplicate.

### 2.9. Co-Immunoprecipitation (Co-IP)

HEK293T cells were transfected with pcDNA3.1 Myc-His-RSK3 and pEGFP-IκBα using Turbofect (Thermo Fisher Scientific Inc, Waltham, MA, USA). Twenty-four hours after transfection, cells were washed with 1X PBS and lysed with 300 μL of RIPA buffer which made by us (2X; 1 M Tris pH7.5, 4 M NaCl, 200 mM EDTA, 10% NP-40) supplemented with a protease and phosphatase inhibitor cocktail mix (Thermo Fisher Scientific Inc., Waltham, MA, USA). Five-hundred micrograms of cell lysate was incubated with a 1:50 dilution of RSK3 and anti-mouse IgG antibodies for overnight. It was then incubated overnight with the protein G agarose beads that were washed four times with PBS. Next, it was washed six times with incubated beads and we made RIPA buffer. The immune complexes were released from the beads by boiling in sample buffer for 5min. Following electrophoresis on 10% SDS-PAGE (Bio-Rad, Hercules, CA, USA), immunoprecipitates were transferred onto PVDF membrane (Bio-Rad, Hercules, CA, USA) and immunoblotted with a specific IκBα antibody (L35A5, Cell Signaling Technology, Beverly, MA, USA).

### 2.10. Immunoblot (IB) Analysis

All cell extracts were harvested in 1X RIPA buffer from homemade solution (2X; 1 M Tris pH7.5, 4 M NaCl, 200 mM EDTA, 10% NP-40), and samples were centrifuged at 13,000 rpm at 4 °C for 30 min. The samples were then boiled in sample loading buffer (Invitrogen, Carlsbad, CA, USA) containing SDS (Sodium Dodecyl Sulphate), and equal amounts of samples were resolved on 10% SDS–PAGE gels, we made, and then transferred onto PVDF membrane (Bio-Rad, Hercules, CA, USA). The membrane was blocked and incubated with the indicated primary antibodies for overnight at 4 °C, and then followed by incubation with horseradish peroxidase (HRP) conjugated secondary antibody (Santa Cruz Biotechnology, Santa Cruz, CA, USA). Proteins were visualized using the enhanced chemiluminescence (ECL) detection system (GE Healthcare Bio-Sciences Corp., Piscataway, NJ, USA). All immunoblot analyses were performed on the ChemiDac™ XRS+ imaging system (Bio-Rad, Hercules, CA, USA). The intensity of each protein band was normalized to that of β-actin to generate the relative intensity.

### 2.11. In Vitro Kinase Assay

For the IκBα kinase assay, active RSK3 and inactive IκBα were incubated in 1X SignalChem kinase assay buffer (SignalChem, Richmond, VA, Canada) with 1 mM DTT (SignalChem, Richmond, VA, Canada) and 1 μM ATP solution (SignalChem, Richmond, VA, Canada). The recombinant active RSK3 (200 ng) protein and its respective substrate IκBα (200 ng) protein were incubated at 30 °C for 30 min and resolved by SDS-PAGE. The proteins were transferred onto PVDF membranes (Bio-Rad, Hercules, CA, USA). The membranes were immunoblotted with the IκBα and phosphor IκBα (Cell Signaling Technology, Beverly, MA, USA) antibodies.

### 2.12. Proliferation Assay

Cell proliferation was examined using a Cell Counting Kit-8 (CCK-8 assay) (Dojindo Laboratories, Kumamoto, Japan), according to the manufacturer’s instructions. Breast cancer cells and normal breast cells were seeded into 96-well plates at 300 cells per well and then incubated for 24 h. The CCK-8 solution was added (10 μL per well), and the plates were incubated for 3 h at 37 °C. The absorbance was detected at 450 nm using a microplate reader (SpectraMax M5, Molecular Devices, Sunnyvale, CA, USA). The experiments were repeated three times independently. MCF10A, MCF7, MDA-MB-231, and MDA-MB-468 cells contained in 10% FBS contained DMEM high glucose were inoculated into 96-well plates at 100 cells per well. After cells were attached on plate, dose-dependent RSK3I (0.1 μM, 0.5 μM, 1 μM, 10 μM) inhibitor was added to the wells and the cells were incubated at 37 °C 12 h to 48 h.

### 2.13. Foci Assay

Foci assays were performed in 12-well plates seeded with MDA-MB-231 cells at ~300 cells per well. After 24 h of seeding the cells, RSK3I was treated dose-differently and DMSO as a control, and then, 14 days later to allow cell growth. After 14 days, the cells were cleaned by PBS and 0.1% crystal violet (MilliporeSigma, St. Louis, MO, USA) in 50% methanol (Fisher Scientific, Bishop Meadow Road, Loughborough, UK) was used to stain for cell counting (Amersco, Solon, OH, USA).

### 2.14. Migration Assay

MDA-MB-231 cells were cultured to 30–50% confluence in 24-well plates. The wounds were made by scratching on the cells using sterile pipette tips. The cells were washed with PBS and then cultured in complete medium for 48 h. The images were captured using a Zeiss inverted wide-field microscope with a Canon G12 camera (Canon, Tokyo, Japan) at the indicated time points.

### 2.15. Apoptosis Assay (FACs Assay)

Apoptosis was evaluated using the annexin V-FITC apoptosis detection kit (Invitrogen, OR, USA, Cat. V13242) according to the manufacturer’s instructions. Breast cancer cells (MCF7, MDA-MB231, and MDA-MB-468) and normal breast cells (MCF10A) were seeded in 6-well plates. RSK3I were treated dose-dependently (Non, Mock (DMSO), 1 μM, and 10 μM) for 24 h. Breast cancer cells were washed once in phosphate-buffered saline (PBS) and resuspended in 100 μL 1X annexin-binding buffer. Samples were analyzed on a FACs (Beckman CytoFLEX, model No. A00-1-1102) (Beckman Coulter Korea Ltd., seoul, Korea) using a flow cytometer with the FITC (Alexa Fluor 488 nm) (Beckman Coulter Korea Ltd., Seoul, Korea) of an argon-ion laser (Beckman Coulter Korea Ltd., Seoul, Korea) for excitation.

### 2.16. Statistical Analysis

All experiments were replicated three times. One-way analysis of variance (ANOVA) was conducted to establish the meaning of variances among the treatment clusters. The t-test was managed for normally scattered. The Newman–Keuls test was used for multi-group comparisons. Statistical significance was defined as * *p* < 0.05 and ** *p* < 0.01.

## 3. Results

### 3.1. RSK3 Interacts with and Phosphorylates IκBα

IκBα, an inhibitor of the transcription factor NF-κB, is degraded when phosphorylated [4], which results in NF-κB activation. To identify IκBα binding partners, we screened a library of about 500 GFP-tagged kinases using the CUPID system [3]. RSK3 was identified as an IκBα binding partner. To confirm IκBα and RSK3 binding, IκBα and RSK3 were cloned into the PKCδ-mRFP and pAC-GFP vectors, respectively, and protein expression was confirmed by Western blot. Bait (PKCδ-mRFP-IκBα) and prey (the GFP-tagged library of ~500 cytosolic kinases) were transfected into HEK293T cells and imaged using confocal microscopy (LSM 710, Carl Zeiss, Germany). PKCδ is translocated to the membrane from the cytosol after PMA treatment. In our screen, after treating with PMA, RFP (IκBα (upper panel) or RSK3 (lower panel)), and GFP (RSK3 (upper panel) or IκBα (lower panel)) translocated to the membrane from the cytosol. The confocal data confirm that IκBα and RSK3 translocated to the membrane from the cytosol and translocation was not influenced by the fluorescent tag used. The profiles obtained from the images were analyzed in the indicated orientations (white lines in Figure 1A). Binding of RSK3 to IκBα was confirmed by co-immunoprecipitation (IP). In these experiments, HEK293T cells were co-transfected with Myc/His-tagged RSK3 and GFP-tagged IκBα; lysates were immunoprecipitated using an anti-RSK3 antibody and immunoblotted using an anti-GFP antibody. Western blot analysis indicated that RSK3 binds directly to IκBα (Figure 1B). To validate this interaction, we performed a MTH assay. Luciferase activity, which indicates RSK3/IκBα binding, was approximately three times higher than that of the pBind-IκBα vector-only control, indicating that RSK3 interacts with IκBα (Figure 1C). Furthermore, IκBα was phosphorylated after the addition of RSK3 in in vitro kinase assays. IKKα was used as a positive control, and RSK3-induced phosphorylation levels of IκBα were higher than IKKα-induced phosphorylation levels of IκBα (Figure 1D). These observations suggest that RSK3 binds directly to IκBα, leading to phosphorylation of IκBα by RSK3. The full western blots can be found at Appendix A.

### 3.2. The CTKD Domain of RSK3 Binds to the N-Terminus of IκBα

The linker region of RSK2 is required for binding to IκBα [11]. However, RSK3 binding to IκBα has not been studied. We constructed IκBα deletion mutants lacking various domains to determine which RSK3 domains are important for binding to IκBα (Figure 2A). We co-transfected cells with WT Myc/his-RSK3 and either WT pAC-GFP-IκBα or one of the deletion constructs. Expression of GFP-IκBα in transfected cells was confirmed using ZOE image and Western blotting (Appendix A). Protein was extracted from the transfected cells using RIPA buffer, immunoprecipitated with an anti-RSK3 antibody, and immunoblotted using anti-GFP and anti-RSK3 antibodies. IP was not observed in anti-RSK3 immunoprecipitations using the IκBα deletion mutant 4 (DM4) (Figure 2B). The other three constructs, which include deletions of the RSK3 CTKD, NTKD, and linker regions, were detected by IP (Figure 2C) [17,34]. Cells were co-transfected with WT pAC-GFP-IκBα and WT pAC-GFP-RSK3 or a deletion mutant form. Transfection efficacy and protein expression were confirmed using ZOE imaging and Western blotting, respectively (Appendix A). Immunoprecipitation assays used anti-RSK3 or anti-normal Ig-G, and anti-IκBα antibody was used for the immune blot. IκBα was not immunoprecipitated with the RSK3 DM1 mutant (Figure 2D). Together, these results suggest that the CTKD domain of RSK3 binds to the N-terminal domain of IκBα.

### 3.3. IκBα Is Phosphorylated after Binding to RSK3

To verify RSK3 binding to IκBα and subsequent IκBα phosphorylation at the cellular level, we transfected MDA-MB-231 cells with various doses of a vector containing the *RSK3* gene and found that IκBα phosphorylation increased in a dose-dependent manner (Figure 3A).

NF-κB is the downstream target of IκBα. RSK3 binds to IκBα and increases IκBα phosphorylation (Figure 1). Phosphorylated IκBα is targeted for ubiquitination and subsequent degradation, and free NF-κB is translocated to the nucleus from cytosol and activated. We performed a NF-κB luciferase promoter–reporter assay using RSK3 treatment. Overexpressed RSK3 induced IκBα phosphorylation via binding to IκBα, and NF-κB luciferase promoter activity increased (Figure 3B). This result indicates that RSK3/IκBα binding activates the transcription factor NF-κB. Conversely, cell-based phospho-IκBα ELISA assays indicated that in HEK293T cells, RSK3 triggered IκBα phosphorylation, but RSK3 shRNA reduced IκBα phosphorylation. RSK3 shRNA was used as a negative control (Figure 3C). Suppression of RSK3 expression by RSK3 shRNA was confirmed using Western blot analysis, and type#4 shRNA (AGGTCCTGAAGCGTCAAGGCTATGATGCG) was selected (Appendix A). Next, we assessed the clinical significance of RSK3 in breast cancer. The Human Protein Atlas database (https://www.proteinatlas.org/ENSG00000071242-RPS6KA2/pathology/breast+cancer accessed on 26 February 2021) indicates that upregulation of RSK3 is correlated with poor prognosis in breast cancer patients (Figure 3D). Collectively, these results demonstrate that RSK3/IκBα binding causes IκBα phosphorylation, resulting in NF-κB activation, and suggest that RSK3/IκBα binding could be a target for breast cancer treatment.

### 3.4. RSK3I Inhibits Binding of RSK3 to IκBα

NF-κB is a cell survival factor that is activated by IκBα phosphorylation [27]. We discovered that IκBα was phosphorylated by RSK3 binding; inhibition of the RSK3/IκBα interaction could decrease NF-κB activation, and thereby increase the rate of apoptosis in cancer cells. Therefore, we performed a screen to identify novel inhibitors of RSK3/IκBα binding. An inhibitor of the RSK3/IκBα binding interface was simulated via docking-based virtual screening. A binding model based on the three-dimensional structure of RSK3 was generated at the Daegu-Gyeongbuk Medical Innovation Foundation (DGMIF).

To identify a RSK3/IκBα binding inhibitor (RSK3I) binding site in RSK3, we performed molecular docking analysis. Given the unavailability of a three-dimensional structure of the RSK3 kinase domain, we used the crystal structure of the RSK2 kinase (PDB code 4JG8), which shares 86% sequence identity with RSK3, to create a homology model structure of RSK3 spanning amino acids 415 to 672 (Figure 4A).

Our findings revealed a potential binding pocket for RSK3I located within the C-lobe of the kinase domain of RSK3 (Figure 4A). A molecular docking study was performed to elucidate the interaction between the RSK3I and RSK3 kinase domain. To measure the activity of these compounds, MTH assays and cellular protein translocation-based screening analyses were performed. Cell proliferation assays were performed to determine the non-cytotoxic doses of the compounds and RSK3I was identified (Figure 4B). We conducted a molecular docking study in which RSK3I was modeled into the RSK3/IκBα binding site using the Glide docking tool to further characterize the interactions between RSK3I and RSK3.

We used MTH assays to confirm that RSK3I inhibited RSK3/IκBα binding. HEK293T cells were transfected with pACT-RSK3 and pBIND-IκBα. Luciferase activity decreased with RSK3I concentration in a dose-dependent manner (Figure 4C). In vitro kinase assays using active RSK3 and inactive IκBα also indicated that Ser32 phosphorylation of IκBα decreased in a dose-dependent manner with addition of RSK3I (Figure 4D). Co-immunoprecipitation was performed to confirm that RSK3I directly inhibits the interaction between RSK3 and IκBα. Lysates from HEK293T cells overexpressing full-length RSK3 and IκBα, with or without the addition of RSK3I, were used for immunoprecipitation with anti-RSK3 antibody, and Western blotting was performed using an anti-IκBα antibody. RSK3I prevented RSK3/IκBα co-precipitation in a dose-dependent manner (Figure 4E). Together, these results suggest that RSK3I inhibitor blocks the RSK3/IκBα binding, thereby inhibiting phosphorylation of IκBα at Ser32.

### 3.5. RSK3I Inhibits Tumorigenesis and Increases Apoptosis in Breast Cancer Cells

RSK3I inhibits RSK3/IκBα binding. We used Western blotting to analyze the basal levels of expression of RSK3 and IκBα in a normal breast epithelial cell line and a panel of breast cancer cell lines. The expression level of RSK3 was not significantly different among the four breast cell lines, but the IκBα expression level in MCF7 was higher than in the other cell lines (Figure 5A). We investigated whether RSK3I influenced the proliferation of breast cancer and normal breast epithelial cell lines. We performed MTT assays and found that RSK3I reduced the cell viability of three breast cancer cell lines, but it failed to reduce the viability of normal breast epithelial cells (Figure 5B). Further, inhibition of RSK3/IκBα binding decreased the proliferation of MCF7, MDA-MB-468, and MDA-MB-231 cells in foci formation assays (Figure 5C, Appendix A). MDA-MB 231 is an extremely invasive breast cancer cell line [25]. We performed a wound-healing assay using MDA-MB-231 cells and observed that the migration of MDA-MB-231 cells was decreased by RSK3I in a dose-dependent manner (Figure 5D). The inhibitor increased the apoptosis rate in breast cancer cells, but not in MCF10A cells, in a dose-dependent manner (Figure 5E). Taken together, these findings suggest that RSK3I, an inhibitor of RSK3/IκBα binding, decreases cell growth, proliferation, and migration in breast cancer cells.

## 4. Discussion

PPIs play important roles in diverse cellular functions, such as signal processing and determining cellular phenotypes, and are crucial for interpreting molecular networks [4,35]. Accordingly, small-molecule drugs targeting PPIs have enormous potential in cancer therapy [4]. We previously used the CUPID system to screen for PPIs and inhibitors for use in targeted cancer therapy [3]. In this study, we screened ~500 GFP-tagged kinases and revealed that IκBα directly binds to RSK3.

Members of the RSK family share high sequence identity (75–80%) [7,17,34]. However, each RSK family contains different phosphorylation sites in their NTKD and CTKD [5,6,7]. Moreover, the RSK family members play different roles in cancer [2,9,36]. RSK1 and RSK2 are associated with cell growth, survival, and proliferation [12,37], whereas RSK4 expression is reduced in ovarian cancer, colorectal cancer, acute myeloid leukemia, and breast cancer [18,37]. Loss of *RSK3* gene transcription is observed in a primary ovarian cancer, suggesting that RSK3 is a putative tumor suppressor gene [13]. However, RSK3 is also a tumorigenic protein in breast cancer [16]. RSK3 is involved in the inhibition of cancer, but its mechanism of action is not well understood [2,16].

IκBα inhibits the transcription factor NF-κB. NF-κB is activated and translocated by phosphorylation, ubiquitination, and degradation of IκBα [25,28,38]. The NF-κB heterodimer released from IκBα is translocated from the cytosol to the nucleus [39]. IκBα acts like a tumor suppressor through its control of NF-κB activation [26].

In this study, we identified a novel PPI between RSK3 and IκBα using the CUPID assay system (Figure 1A). IκBα is phosphorylated by RSK3 (Figure 1D). IκBα is a known NF-κB inhibitor [4]. Because NF-κB is a protein required for cell survival, RSK3/IκBα binding, a novel PPI, represents a cancer drug target. RSK3 is a tumor suppressor in ovarian cancer, and RSK3 protein expression increases with the addition of the inhibitor PI3K in breast cancer. Incremental RSK3 expression in breast cancer causes drug resistance [40]; however, its underlying mechanism are still unknown [7]. Therefore, a preferable approach would be to target the novel interaction we identified between RSK3/IκBα. We screened about 100 chemical compounds for inhibition of RSK3/IκBα binding using a MTH assay and identified one compound, RSK3I (Figure 4B). RSK3I inhibition of RSK3/IκBα binding was confirmed by immunoprecipitation, an in vitro kinase assay, and MTH analysis (Figure 4C–E). Our data indicate that RSK3I directly inhibits RSK3/IκBα binding. RSK3I also decreases cell survival, migration, and proliferation in breast cancer cells (Figure 5B–E), providing new possibilities for cancer therapy. To confirm these effects, it is necessary to perform in vivo testing in preclinical trials. RSK3/IκBα binding represents a potential therapeutic PPI target for the treatment of breast cancer.

## 5. Conclusions

Breast cancer is the most common carcinoma and the leading cause of death from cancer among women worldwide. IκBα regulates NF-κB, which is a tumorigenic protein. We screened a library using the CUPID system to identify IκBα binding partners. Consequently, we identified a novel PPI between RSK3 and IκBα. RSK3 increases IκBα phosphorylation and promotes cell survival. We also identified an inhibitor of the RSK3/IκBα interaction using MTH assay screening. This PPI inhibitor, RSK3I, decreases cell survival, proliferation, and migration of breast cancer cells.

## Figures and Tables

**Figure 1 cancers-13-02973-f001:**
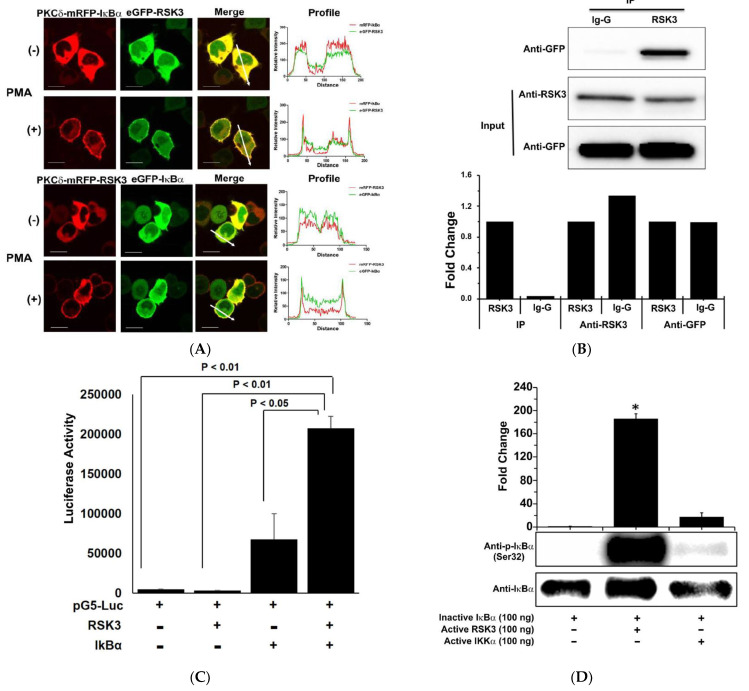
RSK3 interacts with IκBα and regulates its phosphorylation. (**A**) Two pairs of vectors, pKCδ-mRFP-IκBα/pAC-GFP-RSK3 or pKCδ-mRFP-RSK3/pAC-GFP-IκBα, were co-transfected into HEK293T cells, and CUPID analysis was performed using a Zeiss 710 Confocal microscope (LSM 710, Carl Zeiss, Germany) after treating with PMA (0.1 nM). The profiles obtained from the images were analyzed in the indicated directions. Scale bar: 10 μm. (**B**) Myc/His-RSK3 and pAC-GFP-IκBα were co-transfected into HEK293T cells, and RSK3 was immunoprecipitated. Co-immunoprecipitated IκBα was identified by Western blotting using a GFP antibody. Immunoprecipitation was performed three times. Representative results from three experiments are shown. (**C**) pACT-RSK3 and pBIND-IκBα plasmids were co-transfected into HEK293T cells, and a MTH assay was performed. Luciferase activity indicates the change in relative luminescence units normalized to the negative control. Statistical significance was determined by analysis of variance (Newman–Keuls test). (**D**) In vitro kinase assays were performed using inactive IκBα (100 ng)/active RSK3 (100 ng) or inactive IkBα (100 ng)/active IKKα (100 ng). Phosphorylated IκBα was detected by Western blot using a phosphospecific (Ser32) IκBα antibody. Representative results from three experiments are shown.

**Figure 2 cancers-13-02973-f002:**
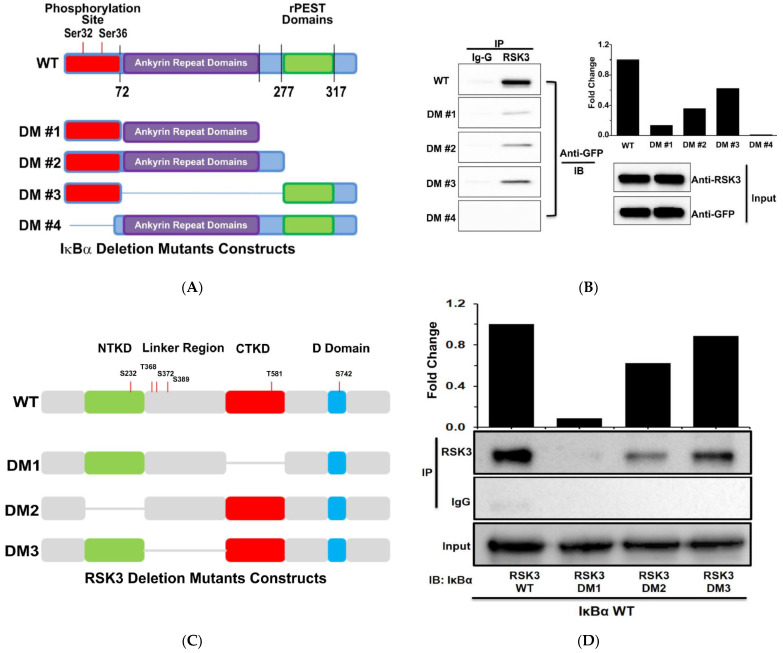
The CTKD domain of RSK3 binds to the N-terminus of IκBα. (**A**) Schematics of full-length and deletion mutants (DMs) of IκBα. (**B**) Full-length Myc/His-RSK3 and WT pAC-GFP-IκBα WT or DM#1–4 were co-transfected into HEK293T cells. RSK3 was immunoprecipitated, and co-immunoprecipitated IκBα was identified by Western blotting using an anti-GFP antibody. Representative results from two experiments are shown. (**C**) Schematics of full-length and DMs of RSK3. (**D**) Full-length WT Myc/His-IκBα and pAC-GFP-RSK3 WT or DM1–3 were co-transfected into HEK293T cells. RSK3 was immunoprecipitated, and co-immunoprecipitated IκBα was detected by Western blotting with an anti-IκBα antibody. Representative results from three experiments are shown. (** *p* < 0.01 and * *p* < 0.05 vs. WT RSK3).

**Figure 3 cancers-13-02973-f003:**
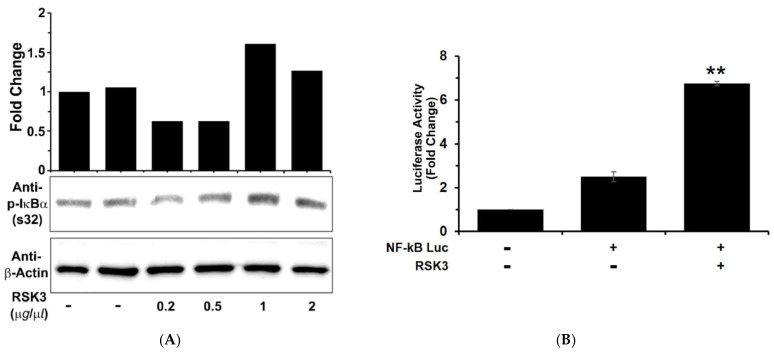
RSK3 activates NF-κB via phosphorylation of IκBα. (**A**) MDA-MB-231 cells were transfected with RSK3, and RSK-induced IκBα phosphorylation was analyzed by Western blotting using a p-IκBα (S32) antibody. Representative results from three experiments are shown. (**B**) NF-κB luciferase activity increased with expression of RSK3. HEK293T cells were co-transfected with 0.5 μg pGL3-NF-κB-Luc, 0.5 μg pcDNA3.1-RSK3, and 0.2 μg pCMV-β-gal. Luciferase activity was normalized against β-galactosidase activity (** *p* < 0.01 vs. control). (**C**) IκBα activation assay. HEK293T cells were inoculated onto a black 96-well plate. Cells were transfected with 100 ng of a RSK3 expression vector alone, the RSK vector and either a control or shRNA (AGGTCCTGAAGCGTCAAGGCTATGATGCG) targeting RSK3 or shRNA alone. IκBα kinetic activity assays were performed (** *p* < 0.01 and * *p* < 0.05 vs. control). (**D**) Kaplan–Meier curve shows that breast cancer patients with low RSK3 expression (*n* = 208) have a better prognosis than those with high RSK3 expression (*n* = 663) (Log rank *p*-value I = 0.018).

**Figure 4 cancers-13-02973-f004:**
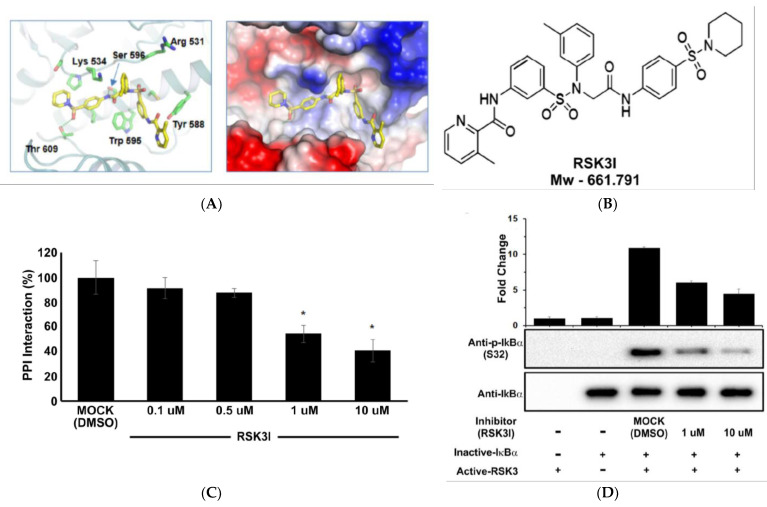
RSK3I inhibits the binding of RSK3 to IκBα. (**A**) A three-dimensional model of the RSK3 structure and the RSK3/IκBα binding inhibitor RSK3I. (**B**) Chemical structure of RSK3I. The molecular weight is 661.791 Da. (**C**) MTH assays were performed by transfecting pACT-RSK3 and pBIND-IκBα into HEK293T cells. RSK3I was added 4 h after transfection in a dose-dependent manner. Luciferase activity was measured 24 h after transfection (* *p* < 0.05 relative to mock). (**D**) RSK3 in vitro kinase assays were performed with dose-dependent treatment of RSK3I (1 μM and 10 μM). IκBα phosphorylation was detected using Western blot analysis. Representative results from two experiments are shown. (**E**) Immunoprecipitation used RSK3 or control Ig-G antibodies in HEK293T cells transfected with Myc/His-IκBα and pAC-GFP-RSK3. RSK3I was added in a dose-dependent manner (0.1 μM, 0.5 μM, 1 μM, and 10 μM). DMSO was used as a vehicle control (MOCK). Co-immunoprecipitated IκBα was identified by Western blotting with an IκBα antibody (* *p* < 0.05). Representative results from three experiments are shown.

**Figure 5 cancers-13-02973-f005:**
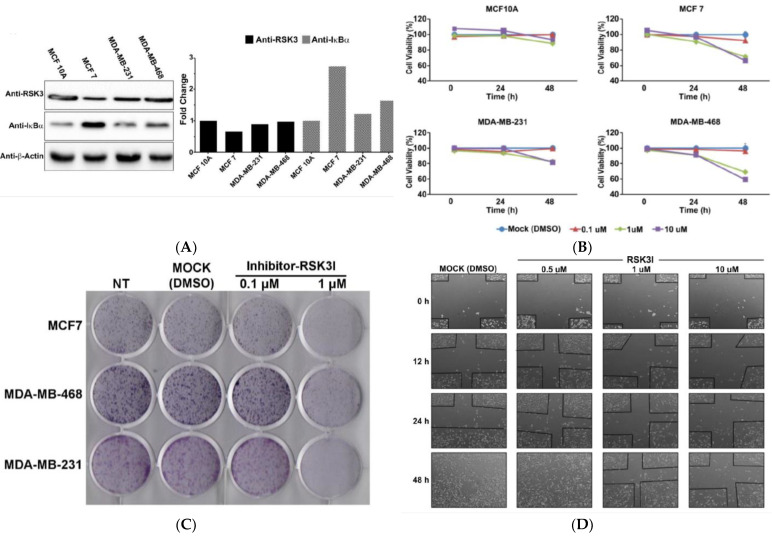
RSK3I inhibits tumorigenesis and increases apoptosis in breast cancer cells. (**A**) The basal expression levels of RSK3 and IκBα in breast cancer and normal breast epithelial cell lines were analyzed using Western blots. Representative results from two experiments are shown. (**B**) Cell viability was analyzed using a CCK assay. RSK3I was added at concentrations of 0.1 μM, 1 μM, and 10 μM for 24 h. (**C**) Foci assays were performed to analyze the growth of breast cancer cells (MCF7, MDA-MB-468, and MDA-MB231). Cells were incubated with RSK3I for 14 days and stained with 0.5% crystal violet. Cell counting was performed using the ImageJ (Java-based image processing program) tool. (**D**) Wound-healing assays were performed to analyze the effect of RSK3I on the migration of MDA-MB-231 cells. The cells were treated with several concentrations (0.1 μM, 1 μM, and 10 μM) of RSK3I, and cell migration was monitored for 48 h. (**E**) The effect of RSKI on apoptosis was assessed by Annexin V/PI staining followed by flow cytometry analysis. RSK3I was added to breast cancer cells (MCF7, MDA-MB-468, and MDA-MB-231) or a normal breast epithelial cell (MCF10A) at concentrations of 1 μM and 10 μM for 24 h, and DMSO was used as a control.

## Data Availability

The data that support the findings of this study are available from the corresponding author, e-mail and H.P, upon reasonable request.

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
