# Peer review of "A Novel Protein–Protein Interaction between RSK3 and IκBα and a New Binding Inhibitor That Suppresses Breast Cancer Tumorigenesis"

_cancers, 2021, doi:10.3390/cancers13122973_

Round 1

Reviewer 1 Report

I recommend publishing as it is. 

Author Response

Thank you so much for your suggestion.

Reviewer 2 Report

  • Full form of NTKD and CTKD in line 58 needs to be introduced and short form can be kept conistent throughout the rest of the manuscript.
  • Line 66 and 67 is unclear.
  • What is FMK in line 69.
  • Number of samples for each experiment needs to be mentioned.
  • In figure 1 PKCdelta translocation to membrane should be shown by including a membrane marker.
  • Figures are of poor quality and skewed

Reviewer 3 Report

In this paper “A novel protein–protein interaction between RSK3 and IκBα and a new binding inhibitor that suppresses breast cancer tumorigenesis”, Yoon and colleagues identified RSK3 (RPS6KA2) as a novel binding partner of IκBα. RSK3 induced phosphorylation of IκBα as well as NF-κB activation. Using an inhibitor of RSK3/IκBα binding identified from a chemical screening approach, they showed impaired RSK3-mediated IκBα phosphorylation reduced survival, proliferation, and migration in breast cancer cells. Although this work provides some interesting observations, the data presented is not solid enough to back up the conclusion.

  1. All Western blots, including expression and phosphorylation level, should be quantified and described how many replicates have been performed in the legends.
  2. Figure 1A: It is really hard to tell the differences between PMA- and PMA+ groups. Both showed co-localization of IkB and RSK3 in cytosol and membrane. The IF images should be quantified to show how many cells had co-localized IkB and RSK3 in the cytosol vs. membrane.
  3. Figure 1B: Was the IP performed using cells treated with PMA?
  4. Figure 1C: Why IkB alone induced luciferase activity of the mammalian two-hybrid assay?
  5. Figure 2: The interaction between IkB and RSK3 shown in Figure 2 were not done using in vitro assays. These were results from co-transfection in 293T cells. Thus, the authors cannot rule out the possibility that other proteins mediate IkB-RSK interaction, unless in vitro interaction assays such as GST-pull down is performed.
  6. Figure 3B: The experiment and description in the main text of this data are confusing. According to the experiment, NFkB promoter-Luc was used. But in the text, the authors described it as NFkB “activity”.  If this is a test of NFkB activity, an NFkB downstream target promoter should be used.  With the current experiment, they can only claim that RSK3 increased NFkB promoter activity, which means NFkB expression level can be induced by RSK3.
  7. Figure 3C: a RSK-/shRSK+ control is missing. Also, how this IkBa kinetic activity assay was done? Did the authors measure the phosphorylation of IkB? If so, Western blot should be shown as well.
  8. Figure 4D, 4E: again, the bands need to be quantified and how many repeats should be indicated.
  9. Figure 5A: The western blot showed that MCF7 expressed high levels of both IkB and RSK3. Other cell lines expressed high RSK3 but low IkB. The description in the main text was incorrect (line 398) which stated that the levels of these proteins were “similar” in the cell lines examined.
  10. Figure 5C, 5D: Since MCF7 expressed high levels of RSK3 and IkB, it should be the most appropriate cell line for testing RSK3/IκBα binding inhibitor. Why the authors used MB231?

Round 2

Reviewer 3 Report

I have no further comments.